# A hybrid feature extraction framework combining PCA and mutual information for gene expression based lung cancer classification

Syed Naseer Ahmad Shah[1]*, Kaartik Issar[2], Rafat Parveen[1]

**1** Department of Computer Science, Jamia Millia Islamia, New Delhi, India, **2** Department of Computer Science, University of Toronto, Canada

* syeddnaseer@gmail.com

## Abstract

Lung cancer remains a leading cause of cancer-related mortality worldwide, with early and accurate diagnosis posing a critical challenge for improving patient outcomes. Gene expression data provide crucial insights for lung cancer classification by revealing underlying biological mechanisms. However, the high dimensionality of such data presents challenges, including computational complexity and overfitting risks. This study proposes a hybrid feature extraction framework combining Principal Component Analysis (PCA) and Mutual Information (MI) to address these issues. PCA reduces dimensionality by capturing key variance patterns, while MI selects features highly relevant to the target class, ensuring an informative and concise feature set. Gene expression datasets from The Cancer Genome Atlas (TCGA) and the International Cancer Genome Consortium (ICGC) were integrated, focusing on common genes. The hybrid PCA-MI framework was applied to rank genes, and the selected features were used to train a Convolutional Neural Network (CNN) for lung cancer classification. The genes ranked by the hybrid model were further analysed using protein-protein interaction (PPI) networks to identify hub genes, enhancing biological interpretability. The proposed framework was benchmarked against ten other feature extraction methods, including Lasso, Random Forest, Autoencoder, and PCA alone. The CNN classifier achieved superior performance with the PCA-MI features, attaining 98% accuracy and 98% precision. Training and validation curves demonstrated stable learning behaviour, and confusion matrix analysis confirmed robust predictions. Hub gene identification through PPI analysis validated the biological significance of the ranked genes. This study presents a robust framework for lung cancer classification by leveraging the strengths of PCA and MI, integrating deep learning and PPI analysis to address high-dimensional data challenges, and setting a foundation for future research in multi-omics data integration and enhanced diagnostic strategies.

**Data availability statement:** The data and code that support the findings of this study are available on https://github.com/SyeddNaseer/PCA_MI_GeneFeatureExtraction.

**Funding:** The author(s) received no specific funding for this work.

**Competing interests:** The authors have declared that no competing interests exist.

**Abbreviations:** PCA, Principal Component Analysis; MI, Mutual Information; TCGA, The Cancer Genome Atlas; ICGC, International Cancer Genome Consortium; CNN, Convolutional Neural Network; NSCLC, Non-Small Cell Lung Cancer; SCLC, Small Cell Lung Cancer; PPI, protein-protein interaction; MSE, Mean Squared Error; Lasso, Least Absolute Shrinkage and Selection Operator; ANOVA, Analysis of Variance.

## 1. Introduction

Lung cancer is one of the most common as well as deadliest cancers in the world, and about 1.8 million people die annually due to this cause, accounting for nearly 18% of all deaths from cancer. It is also one of the cancers with the highest number of diagnoses annually and is quite rampant in high-income as well as low-income countries [1]. In general, Lung cancer is primarily classified into two major types: Small Cell Lung Cancer (SCLC) and Non-Small Cell Lung Cancer (NSCLC). Each category has different biological features, treatment responses, and survival traits, so appropriate classification is crucial in dictating the process of proper treatment planning and favourable patient outcomes. NSCLC constitutes about 85% of all lung cancer, and it further can be categorised into three main classes based on the difference in their histopathological and molecular profiles: Adenocarcinoma It constitutes around 40% of lung cancer and is the most common type of NSCLC [2]. It commonly occurs at the periphery of the lung and, more commonly, in nonsmokers and younger patients. Adenocarcinoma grows relatively slowly compared to other carcinoma forms, and histological exams often show glandular structures. Squamous Cell Carcinoma accounts for approximately 25–30% of NSCLC cases; squamous cell carcinoma usually develops in the central regions of the lungs, most often in the larger bronchi [3]. It is strongly associated with a history of smoking and typically exhibits keratinisation and intercellular bridges in histopathological examination. Squamous cell carcinoma is a more aggressive tumour and typically invades the surrounding tissues aggressively. Large Cell Carcinoma accounts for approximately 10–15% of NSCLC. Large cell carcinoma appears in any lung location as a form of solid mass and is generally diagnosed once the disease is advanced [4]. In this category, large undifferentiated cells that do not resemble either adenocarcinoma or squamous cell carcinoma are seen. Large cell carcinoma is particularly aggressive, and the tendency to metastasise early is relatively high. The three major types of Non-Small Cell Lung Cancer (NSCLC), Adenocarcinoma, Squamous Cell Carcinoma, and Large Cell Carcinoma, are shown in **Fig 1**, illustrating their different growth patterns and histopathological features. SCLC accounts for approximately 15% of lung cancer diagnoses. The growth rate in SCLC is assertive, with rapid spread to distant sites.

It derives from neuroendocrine cells within the epithelium of the bronchial tree and occurs almost exclusively in the history of heavy smoking. The cells are small, round, and densely packed, hence the term "oat cell" carcinoma in histological terms. The aggressive nature of SCLC can mean different protocols regarding the treatment of SCLC compared to NSCLC and the use of chemotherapy and radiation therapy rather than surgery for resection [5,6]. The two problems, mortality and poor prognosis, have thus necessitated the evolution of a developing diagnostic and classification tool for lung cancer. In any case, the traditional classification methods have been based on a histopathological examination. Although effective, these can be supplemented and enhanced by molecular approaches like gene expression analysis [7]. Indeed, by expression profiling, one can study thousands of genes simultaneously and thus obtain a better understanding of the molecular underpinnings of

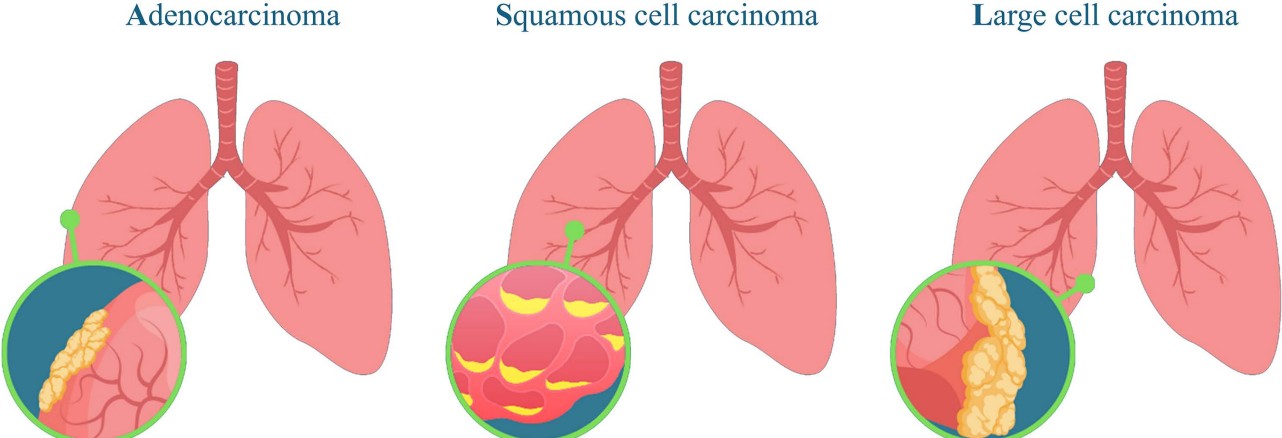

**Adenocarcinoma**  **Squamous cell carcinoma**  **Large cell carcinoma**

**Fig 1. Illustration of different types of lung cancer, highlighting the affected lung areas and unique cellular patterns for each type.**

the subtypes of lung cancer, allowing more accurate classification; it may also inform treatment decisions, identify novel therapeutic targets, and improve early detection strategies through intensive analysis [8]. However, gene expression datasets are incredibly challenging to analyse, primarily because of their high dimensionality. Gene expression profiles for each sample comprise thousands of gene measurements, but the number of samples is much smaller, generally several hundred in cancer studies. This is precisely what occurs in machine learning models due to an imbalanced number of features (genes) versus samples, so the model learns noise and not meaningful patterns and thus can't generalise well to new data. Therefore, successful feature extraction means the reduction of dimensionality without losing informative signals about the biological processes underlying lung cancer [9]. Feature extraction is an essential preprocessing procedure in gene-expression data since it identifies relevant features from raw data and transforms them into a manageable, informative subset. Selecting appropriate features may improve the efficiency of computationally expensive classification algorithms while, at the same time, improving chances of getting better accuracy since focusing on the most relevant data to differentiate between, or to distinguish, the various classes or subtypes of lung cancer. Otherwise, if features are adequately extracted, models can be cluttered with redundant and unimportant features, which may further generate noise, reduce interpretability, and hide meaningful biological patterns [10]. Thus, selecting informative features is crucial in high-dimensional datasets such as gene expression data for improving classification models' reliability and accuracy. A hybrid feature extraction framework proposes PCA and mutual information to overcome dimensionality challenges in lung cancer gene expression data. PCA is a well-established dimensionality reduction technique that converts high-dimensional data into a lower-dimensional space by identifying key features, orthogonal components that capture maximum variance. PCA reduces noise and overall data complexity by retaining only a subset of these principal components. However, it ignores the connection between the features and the target labels; therefore, the features are only sometimes ordered by relevance to the task of classification [11]. We further improve the relevance features through the mutual information method – the technique of mutual dependence on the rank target class. The idea was that mutual information would rank their nonlinear dependency on a target class, selecting features that reveal more perfectly, or best, what distinguishes types of lung cancer. So, PCA and MI use each technique's best features. PCA should reduce dimensionality as much as possible since it captures only the main variance. In contrast, MI selects the feature with the highest relevance, ensuring an informative feature set for classification. The underlying motivation of this hybrid approach will be to optimise lung cancer classification accuracy through the correct balance of variance from PCA and feature relevance from MI [12,13]. This framework is supposed to improve classification accuracy, reduce noise, and enhance the interpretability of

gene expression data. We test the proposed hybrid method with a CNN classifier and compare its performance against models based on individual feature extraction techniques. Further hub genes were identified along with the protein-protein interaction to know the biological relevance. Our results show that combining PCA and MI results in a more robust feature set for classification tasks in lung cancer, which makes hybrid feature extraction methods potentially useful for gene expression analysis in complex diseases like lung cancer.

## 2. Related work

PCA is one of the most essential tools for dimensionality reduction in gene expression studies. It has thousands of gene features but still uses relatively small sample sizes. PCA captures essential variance patterns effectively to reduce complexity. PCA has multiple applications in bioinformatics, mainly for noise removal and visualisation purposes. However, while PCA focuses its attention on variance, it is inherently not concerned with the relevance of features towards the target classification, which is critical in cancer studies. Recent studies further project the relevance of PCA through relevance-based techniques, such as mutual information. It is a technique that chooses the features most informative about the class label, as it balances the variance representation with information relevant to classification [14,15]. The limitation of the single-method-based feature selection has motivated the researchers to explore hybrid approaches. Hybrid approaches combine methods from both filter-based and wrapper-based approaches to address redundancy and noise problems common in gene expression data. Example: Hybrid methods which combined MI with the genetic algorithm were applied in gene selection and optimised to classify cancers whereby the classifiers found to have both precision and high computational efficiency of Hybrid methodology usually outperform any single-method techniques offering concise yet highly informative feature sets [16,17]. It thus enhances classification accuracy and controls overfitting risk. With the advent of deep learning, the hybrid feature selection approach is now making sense in preparing datasets with models like CNNs, which successfully find patterns in a complex dataset. In cancer classification tasks, the performance of these CNNs is high when equipped with a sound feature selection process. Recently, for example, it has been shown that by integrating MI-based feature selection with machine learning models, significant accuracy gains were achieved on gene expression data, mainly because hybrid approaches guarantee the selection of features that do not only reduce their dimensionality but also very discriminative in classification task [15,17]. Comparison studies have shown that hybrid approaches for feature selection, such as PCA-MI, outperformed single-method approaches like Lasso and ElasticNet. Such studies validate the improvement of classification accuracy and stabilisation of the learning behaviour of the model due to PCA-MI hybrids. A comparison of experiments between PCA-MI hybrids and similar techniques, such as Random Forest and ElasticNet, led to significant improvements in precision and accuracy, mainly when implemented along with deep models, namely CNNs [18].

## 3. Materials and methods

The study proposes a hybrid PCA-MI framework for the extraction of features and classification of lung cancer gene expression data. It includes preprocessing of data, dimension reduction using PCA, and feature selection based on mutual information. The reduced feature space is fed to the CNN model and further optimised for the classification. The performance across various feature extraction techniques is compared. A comprehensive PPI network analysis was conducted to identify hub genes. The analysis focuses on the biological significance of interpreting identified genes' functions. The methodology is detailed as shown in **Fig 2**. which provides an overview of the workflow.

### 3.1. Dataset collection and preparation

The paper utilises gene expression data derived from two widely acknowledged public repositories: TCGA and the ICGC. The TCGA dataset comprises 1,153 samples, of which 541 are adenocarcinoma samples, 502 are squamous cell carcinoma samples, and 110 are normal samples. A Python script has been applied to ensure the data can be downloaded without errors. Similarly, we obtained the ICGC dataset of 543 samples, of which 488 belonged to adenocarcinoma

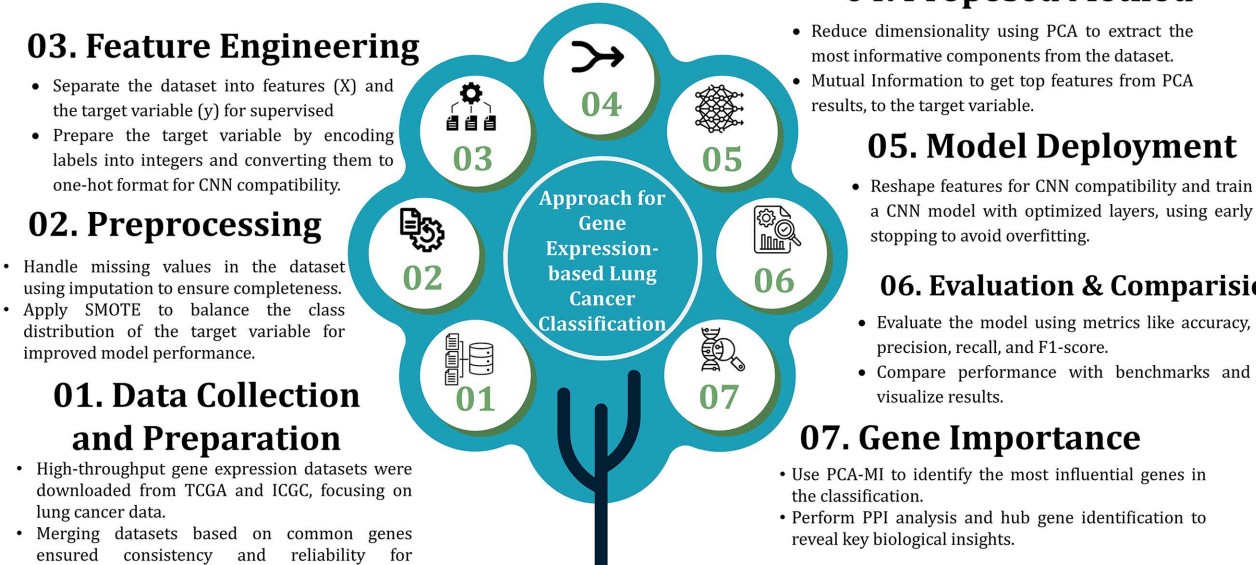

**Fig 2. A Seven-Stage Framework for Gene Expression-based Lung Cancer Classification: From Data Collection to Model Deployment with Integrated Feature Engineering and PCA-MI Feature Selection.**

samples and 55 were normal. These two datasets collected independently provide complementary information required to develop a robust and comprehensive benchmark dataset. [19].

**3.1.1. Data preprocessing.** To prepare the data for analysis, we applied Z-score normalisation to standardise gene expression values. This normalisation ensured consistency across samples from different repositories by transforming each gene expression value into a mean of zero and having a standard deviation of one. The formula for Z-score normalisation [20],

$$Z = \frac{X - \mu}{\sigma}$$

(1)

Where X is the value being normalised, μ is the mean of the dataset, and σ is the standard deviation of the dataset. This step minimised any biases or variations stemming from differences in sequencing platforms or data processing protocols, making the datasets more compatible.

**3.1.2. Merging.** We have combined TCGA and ICGC with their common genes into a single benchmark set after standardisation. This means that only the genes whose expression is reflected in the datasets allow a comparison for both sources. Mean has been used to impute the missing values after merging. For any gene, missing values were replaced by the average expression level for that particular gene across the rest of the samples because it lacked expression values for any given samples. It ensured no gaps within the dataset, thus permitting uninterrupted analysis. [21].

**3.1.3. Class balancing.** The dataset showed class imbalance, specifically with fewer normal samples. Thus, the Synthetic Minority Over-sampling Technique was applied to rectify this problem. SMOTE is a popular technique for synthetic sample generation to balance class distribution. In this study, the SMOTE algorithm generates synthetic samples for the minority class, which are the normal samples. It does so by randomly selecting a minority class sample with one of the nearest neighbouring samples; the algorithm then interpolates these two samples to create the new synthetic sample. [22].

$$x_{new} = x_i + \delta \times (x_{nn} - x_i) \qquad (2)$$

$\delta$ is some random number between 0 and 1; therefore, the generated samples would be different. $x_i$ and $x_{nn}$ are the feature vectors of the minority samples. This created artificial samples that enriched the minority class, thereby balancing the data and improving the classification model's performance on minority classes [23].

**3.1.4. Labelling.** The samples were labelled for classification into their categories: adenocarcinoma, squamous cell carcinoma, and routine. This labelling provided an organised foundation for analysis, both in the tumour vs standard classification and the finer subtype classification within lung cancer. Therefore, these labelled and pre-processed data furnished a solid foundation for downstream machine-learning tasks and model development. [24].

### 3.2. Problem of high dimensionality and the need for feature extraction

The gene-expression profiles have high dimensionality, with thousands of features (genes) and very few samples for analysis. This causes an inequality problem, which has several issues. The risk of overfitting, where the model learns noise rather than the underlying patterns to poor generalisation in case new data sets are presented. Moreover, high dimensionality makes computations inefficient and makes it difficult to train some models properly. [25]. To overcome these problems, our study has used a hybrid feature extraction technique using PCA and mutual information with only helpful features retained with noise reduction and prevention of computational load in the process.

### 3.3. Proposed hybrid feature extraction method

**3.3.1. Principal component analysis.** PCA is a high-dimensional data dimensionality reduction technique that brings any dataset originating in a high-dimensional space into a lower-dimensional space and allows one to identify principal components, which are the directions holding the maximum variance within the data. Working on these principal components, PCA effectively decreases the dimensionality of the dataset. At the same time, noise becomes minimised, and it gets to be computationally more efficient without somehow sacrificing the structure of the original data. In this context, PCA is used on the gene expression profiles to keep only those components retaining over 95% of the total variance from the dataset [26]. These principal components are essentially uncorrelated linear combinations of the original features that help reduce redundancy and improve the dataset's general quality by filtering out less informative variables. Following PCA execution, the components were ranked systematically according to their respective contribution to the overall variance, allowing us to choose only the top-ranked components for further analysis. Thus, carefulness in the process ensured that the most informative aspects of the gene expression data were preserved, which provides a robust foundation for subsequent classification tasks [27].

**3.3.2. Mutual information.** MI is one of the key metrics in the analysis of gene expression data that quantify the dependency between any individual features and the target variable, which, in the context of this study, are the class labels assigned to different types of lung cancer. MI allows for a process of relevance-based selection that would make the features most likely to describe the gene expression profile shared with the class label selected for classification. [28,29]. This is very relevant in classifying lung cancers because it makes a big difference in identifying the underlying types involved in the diagnosis and strategies in treatment that should be followed. In this work, the MI was computed for each transformed component using PCA and corresponding class labels of lung cancer. The analysis focused on finding components that led to the highest MI scores, as these components were considered to have the highest predictive capability towards classifying the various types of lung cancer. This ensures that the feature set resulting from such a procedure is not only representative of an underlying data structure but highly relevant for accurately predicting classification in lung cancer cases and, thus, enhances the overall effectiveness of a classification model [30].

**3.3.3. Integration of PCA and MI.** The components preserved from PCA were ranked further by the MI scores that they achieved, which is an essential step in moving toward further tuning the features toward classification. Thus, what

was demonstrated here is the merit of bringing together dimensionality reduction through PCA and relevance-based feature selection through MI. Firstly, it reduces the dimensionality of the high-dimensional gene expression dataset and converts the original features into a reduced set of uncorrelated principal components. The reduction not only aided in minimising the complexities attached to the dataset but also helped filter out noise and redundancy, thus conserving the critical structure of data and capturing the significant variance. Next, the retained components' MI scores, a measure of component dependency on the class labels for lung cancer, were considered [31]. Then, an analysis of the ranking of the components by their MI scores was carried out to highlight the principal components that better informed the classification problem at hand. Only the most highly ranked component, the features whose MI scores are highest, have been selected for the final feature set. This results in a compact feature set after dimensionality reduction in PCA and is highly relevant to classifying lung cancer subtypes. Essentially, the integration of PCA and MI incorporates a more robust feature set, which would better optimise the ability of the model to classify the various classes of lung cancer while retaining the most informative characteristics of the underlying data. [32].

### 3.4. CNN classifier for lung cancer classification

CNN has become a vital tool for examining gene expression data with complex assignments such as classifying lung cancer. The gene-expression profiles are generally of a vast number of genes, hence high-dimensional: this gives a bad warning as it is known to impose overfitting and lacks clear interpretability. A CNN addresses both issues by automatically learning to extract essential features from convolutional layers while capturing intricate relationships between genes. This method is helpful because it does not demand extensive manual feature engineering; the CNN can instead identify patterns in the data that may indicate disease-specific gene-expression signatures [33,34]. For example, convolutional layers of the CNN recognize expression patterns, where pooling layers downsample data to emphasise critical features and reduce complexity, thus making analysis efficient yet precise. The CNN also uses non-linear activation functions like ReLU, thereby capturing complex and highly non-linear relationships between genes, often the states' markers. Dropout layers help prevent over-fitting by randomly turning off some neurons whilst training, making the model more robust and thus better generalizing on unseen data. Its architecture consists of convolutional, pooling, and dense layers that provide an all-inclusive representation of high-dimensional gene data for multiclass classification in lung cancer, and it can output the probabilities across various types of cancers, distinguishing between the several subtypes of lung cancer based on an understanding of even subtle distinctions in patterns of gene expression. The strength of the CNNs applied to a feature-reduced data set—for instance, hybrid methods such as PCA and mutual information filtering could be used—is to focus only on the features of biologically highest importance, thus enhancing classification accuracy. Lung cancer classification CNNs provide an effective instrument when traditional methods fail, eventually discovering key biomarkers and pathways that would be useful in precision medicine and developing novel therapeutic strategies [35].

### 3.5. CNN model architecture

The proposed CNN architecture is designed to classify lung cancer subtypes effectively, as shown in **Fig 3**. It integrates hybrid PCA-MI-based feature extraction with a multi-layered neural network structure to capture intricate patterns in the data. The architecture comprises input, convolutional, pooling, fully connected, and output layers, optimised through extensive hyperparameter tuning and evaluated using robust performance metrics.

*Input Layer:* The input layer of CNN was designed in such a way that it will accept the final feature set produced by hybrid PCA-MI. Every sample was taken as a feature vector consisting of the selected components.

*Convolutional Layers:* The architecture consists of several convolution layers operating at different filter sizes to capture different patterns from the feature set. A ReLU activation function is applied for every convolutional layer so that non-linearity can be injected into the model. Max-pooling layers follow convolutional layers to downsample feature maps that reduce both the dimensionality and computation required for the forward and backward pass.

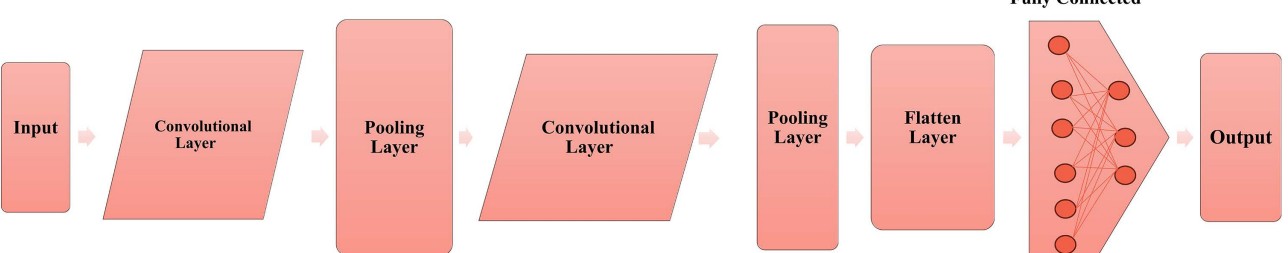

**Fig 3. A block diagram of the CNN architecture shows the flow from the input layer through convolutional pooling and fully connected layers to the output layer to classify the samples as either lung cancer or not.**

*Fully Connected Layers:* The fully connected layers presented after the convolutional layers have been added to include features extracted from the preceding layers and to make predictions related to the learned patterns. Dropout layers were used to prevent overfitting. The dropout rate is between 0.3 and 0.5 [33,36].

*Output Layer:* It is the final layer of the network-softmax layer. Units equal to the lung cancer classes like (adenocarcinoma, squamous cell carcinoma, large cell carcinoma, and small cell lung cancer). This layer represented class probabilities for each sample.

**3.5.1. Training parameters.** The Adam optimiser is used with an initial learning rate of 0.001. Categorical cross-entropy loss measures the difference between predicted and known class labels. Early stopping based on validation loss was performed to avoid overfitting at train time [37].

**3.5.2. Hyperparameter tuning.** Key hyperparameters, including the number of convolutional layers, filter sizes, learning rate, batch size, and dropout rate, were tuned using grid search accompanied by cross-validation. This iterative process helped identify the best model configuration.

**3.5.3. Evaluation metrics.** The model's performance was evaluated with a range of metrics, such as accuracy, precision, recall, F1-score, and the area under the ROC curve (AUC). These metrics comprehensively assessed the classification performance across different lung cancer classifications [38].

## 3.6. Comparative analysis with individual feature extraction techniques

We independently compared our proposed PCA-MI hybrid feature extraction method against ten established techniques applied to the pre-processed dataset. These techniques provide other means of dimensionality reduction and feature selection, giving insight into how variations in methods impact CNN performance in classifying lung cancer from gene expression data.

**3.6.1. AutoEncoder.** An autoencoder is a feedforward deep architecture-type neural network specially designed for unsupervised learning. It aims to discover compressed and meaningful information. This structure makes autoencoders useful for dimensionality reduction and feature extraction since they capture crucial information within the data without irrelevant details and noise [39]. Two primary stages work in the autoencoder: encoding and decoding. The encoding process compresses input data into a lower dimension latent representation called "bottleneck," which captures the core informative aspects of the input. Mathematically, the encoder transforms input x through the function.

$$z = f(x) = \sigma\ (W_e x\ +\ b_e)$$

(3)

$W_e$ is the encoder's weight matrix, $b_e$ is the bias, and $\sigma$ is a non-linear activation function commonly chosen as ReLU or sigmoid. In the decoding stage, the network reconstructs the input data from the compressed representation. The decoder function models this process.

$$\hat{x} = g(z) = \sigma(W_d z + b_d) \tag{4}$$

$W_d$ and $b_d$ are the decoder's weight matrix and bias term, respectively. The output of the decoder, $\hat{x}$, represents the reconstruction of the original input $x$, to make $\hat{x}$ as close as possible to $x$ by minimising the reconstruction error. This task's most common loss function is Mean Squared Error (MSE), which quantifies the difference between each input feature and its reconstructed value [40,41].

$$L(x, \hat{x}) = ||x - \hat{x}||^2 = \sum_{i=1}^{n}(x_i - \hat{x}_i)^2 \tag{5}$$

Where n is the number of features, the autoencoder learns its parameters at training time to minimise this loss. This enables it to learn compact and informative input representations, which is why autoencoders are so good at analysing complicated data like gene expression profiles that contain underlying nonlinear relationships critical to understanding the interaction of genes and significant biological patterns, reducing noise in such data.

   **3.6.2. PCA.** PCA is a very commonly applied linear dimension reduction method. It transforms the high-dimensional data into a smaller a collection of uncorrelated variables known as principal components. The principal components are ordered combinations of the features in terms of variance that explain the data set and are linear combinations of the original features. The first principal component captures the direction of maximum variance, followed by further components, each of which captures the most significant possible variance in the remaining orthogonal directions [42]. This is why PCA captures the most meaningful variance in data with only a few principal components, thus reducing its dimension and capturing the most significant patterns. Mathematically, PCA is done by calculating the covariance matrix of the data and then finding its eigenvectors (principal components) along with their eigenvalues. Eigenvectors are directions of maximum variance, and eigenvalues show the magnitude of variance in those directions. If X represents the original matrix of n samples and p features, then the covariance matrix C can be calculated as:

$$C = \frac{1}{n-1} X^T X \tag{6}$$

where $X^T$ is the transpose of $X$. The eigenvectors of $C$, denoted $v_1, v_2, \ldots, v_k$ are the principal components, and the associated represent the variance each component explains. Sorting the eigenvectors in descending order of their eigenvalues $\lambda_1, \lambda_2, \ldots, \lambda_k$ enables us to retain only the components with the highest variance, thereby achieving dimensionality reduction. For gene expression data, PCA is beneficial, as it reduces the large feature space to a manageable size, capturing the main variations in gene expression patterns while filtering out noise and less relevant fluctuations [43,44]. This reduction simplifies data processing and improves the efficacy and performance of downstream machine learning models by focusing on the most informative aspects of the data.

   **3.6.3. Mutual Information.** MI is a statistical method that quantifies the dependency between two variables, providing insights into how much knowing one variable reduces uncertainty about the other. It is precious in feature selection, as it helps determine the relevance of each feature about a target class by measuring how much information the feature contributes toward predicting the target. In the context of gene expression data, MI is beneficial because certain genes have strong associations with specific cancer types, making it essential to identify those genes that carry the most predictive information [45]. MI between two discrete random variables X (e.g., a gene's expression level) and Y (e.g., cancer type) is defined as:

$$I(X; Y) = \sum_{x \in X} \sum_{y \in Y} p(x, y) \log \frac{p(x, y)}{p(x)p(y)} \tag{7}$$

$p(x, y)$ represents the joint probability distribution of X and Y., $p(x)$ and p(x) and $p(y)$ are the marginal probability distributions of X and Y, respectively. The MI score $I(X; Y)$ is higher when X and Y have a strong dependency, meaning that knowing the value of X significantly reduces uncertainty about Y and vice versa. In feature selection for classification, this score can rank features by their relevance to the target class, allowing us to prioritise features (genes) that provide the most discriminative information for classification tasks [46,47]. Focusing on features with high MI scores can improve model performance by selecting a subset of features that maximise information gain, which is particularly advantageous when dealing with high-dimensional gene expression data.

**3.6.4. AutoEncoder and Mutual Information.** The hybrid approach is a vital feature selection technique that harnesses AutoEncoders' strengths with mutual information to create a concise yet highly informative feature set. This approach is beneficial in applications such as gene expression analysis, where the dimensionality is high, and choosing the most informative features. plays a crucial role in effective classification. The autoencoder is an artificial neural network wherein the training for unsupervised learning happens; hence, it compresses the information in the data by learning a compressed representation. It has two pieces: an encoder, which translates this high-dimensional input to a lower-dimension latent space and captures the essence of the structure in the data, and a decoder, which reconstructs the original input from this compressed form. Autoencoder filters out noise in the data, thus preserving the main patterns, and it results in a condensed representation that retains all the essential information provided by the original data set by minimising reconstruction error [48]. Once compressed, mutual information is utilised to decide the crucial features along this compressed representation of the target variable. Mutual information measures the dependency between the values of two variables; that is, how much knowing the value of one reduces the uncertainty in the value of the other. In such a scenario, MI scores help pick the most relevant features in the compressed representation for predicting the target class. The features have higher MI scores, so the low MI features are discarded to yield a compact, predictive representation, ending with the final list of features. AE combined MI by combining unsupervised feature reduction in auto-encoders with supervised feature relevance estimation using mutual information [49]. Then, a feature set that captures the critical data patterns is optimised for the specific classification tasks, improving the efficiency and accuracy of subsequent machine learning models. This hybrid approach is beneficial for challenging datasets, like gene expression data, where discovering a targeted subset of informative genes may lead to better classification and understanding of underlying biological patterns.

**3.6.5. Lasso (Least Absolute Shrinkage and Selection Operator).** Lasso is one of the linear regression methods incorporating L1 regularisation that enforces sparsity on the model's coefficients. Thus, it is essential as an application for feature selection. Unlike ordinary linear regression, which fits the model by minimising the residual sum of squares, Lasso regression incorporates a penalty based on the absolute sum of the coefficients so that some become zero [50]. This is one of Lasso's properties. Using gene expression data, for instance, many features may be distant from having at least a weak relationship with the target variable, for example, a particular class of disease. Lasso, thereby reducing the dimension of the dataset, shrinks the coefficients of irrelevant or redundant features to zero. It retains those features which have the most potent predictive power.

$$Loss = \frac{1}{2n} \sum_{i=1}^{n} (y_i - \hat{y}_i)^2 + \alpha \sum_{j=1}^{p} |\beta_j|$$

(8)

the first term $\frac{1}{2n} \sum_{i=1}^{n} (y_i - \hat{y}_i)^2$ is the Mean Squared Error (MSE) between the observed target values $y_i$ and the predicted values $\hat{y}_i$, where $n$ is the number of samples. The second term $\alpha \sum_{j=1}^{p} |\beta_j|$ represents the L1 regularisation penalty, with α\alphaα being a regularisation parameter that determines the strength of this penalty and $\beta_j$ denoting the coefficients of the features. When $\alpha$ is set to zero, Lasso regression behaves like ordinary linear regression without any penalty [51]. However, as $\alpha$ increases the penalty term grows, leading to more coefficients being shrunk toward zero. This process

continues until only the most significant features—those with the largest effects on the target variable—retain non-zero coefficients. The choice of $\alpha$ is thus crucial: a smaller $\alpha$ allows more features to be retained, while a larger α forces more coefficients to zero, leaving only the most relevant features. Cross-validation often determines This tuning parameter to achieve optimal feature selection and predictive accuracy. By reducing the number of active (non-zero) coefficients, Lasso regression simplifies the model and enhances interpretability, as it isolates features that contribute most meaningfully to the model's predictions [52,53]. This makes Lasso regression particularly advantageous for gene expression data analysis, where thousands of genes may be analysed, yet only a subset might be truly relevant to the classification or prediction task. Through its regularisation mechanism, lasso regression aids in identifying these essential genes, making it a highly effective approach for dimensionality reduction in genomics and other fields with high-dimensional data.

### 3.6.6. Random forest.

Forest is an ensemble learning technique that uses the outputs of multiple decision trees to produce a more robust and accurate prediction. Compared to a single decision tree, which is bound to overfit and becomes sensitive to data variations, Random Forest creates a "forest" of individual diverse decision trees, each trained on a randomly sampled subset of the original data and a random selection of features. Random Forest reduces variance and provides better generalising ability when it takes the average of the predictions of the individual trees. It is useful in high-dimensional and complex datasets. In a Random Forest, every decision tree is built recursively using feature values to create a random split at each node that will form the branches up to the predictions in the leaf nodes [54]. In this stage, the model picks those splits that lead to minimal impurity or disorder of the target variable. Probably the most popular measures of impurity are Gini impurity and entropy. This model would have assigned higher importance scores to features whose contribution significantly reduces node impurity for all the trees. Then, the importance of each feature is estimated as the average reduction in impurity caused by splits on that feature over all the trees in the forest. The process for computing the feature importance for Random Forests involves aggregating the decrease in impurity from all splits on a feature across all trees. If a feature $X_j$ splits a node t in tree T, the decrease in impurity for that split is defined as:

$$\Delta I(t, \, X_j) = I(t) - \left( \left( \frac{n_{left}}{n} \, I(t_{left}) + \, \frac{n_{right}}{n} \, I(t_{right}) \right) \right)$$

(9)

$I(t)$ is the impurity of the node before the split, $I(t_{left})$ and $I(t_{right})$ are the impurities of the left and right child nodes after the split, $n$ is the total number of samples in node $t$, $n_{left}$ and $n_{right}$ are the numbers of samples in the left and right child nodes. The total feature importance score for the feature $X_j$ is obtained by summing the impurity decreases $\Delta I(t, \, X_j)$ over all nodes where $X_j$ was used to split across all the trees in the forest. This ranking of features by importance helps prioritize variables most relevant to the target variable, allowing Random Forest to serve as both a predictive model and a feature selection method. Random Forest is particularly advantageous for complex datasets like gene expression data, where interactions among thousands of genes can influence the outcome [55,56]. By ranking genes based on importance scores, Random Forest helps identify those most associated with the target variable, making it a valuable tool in genomics, where identifying influential genes can aid in understanding biological pathways or diagnosing diseases.

### 3.6.7. ANOVA (Analysis of Variance).

ANOVA is a statistical technique that tests whether there is a significant difference in the means of a feature across multiple classes by comparing the variation between groups to the variation within groups. In gene expression analysis, ANOVA is particularly useful for identifying genes that show distinct differences in expression between groups, such as cancerous and non-cancerous samples. The test calculates two main types of variances: The between-group variance (which measures the difference between group means and the overall mean) and the within-group variance (which reflects the variation of individual observations within each group from their respective group mean) [57]. The formula gives the ratio of these variances, known as the F-ratio.

$$F = \frac{MS_{between}}{MS_{within}}$$

(10)

$$MS_{between} = \frac{\sum_{i=k}^{k} n_i(\overline{X}_i - \overline{X})^2}{k-1} \tag{10.1}$$

$$MS_{within} = \frac{\sum_{i=1}^{k} \sum_{j=1}^{n_i} (\overline{X}_i - \overline{X})^2}{n-k} \tag{10.2}$$

Here, $k$ tells the count of groups $n_i$ represents the number of observations in group $i$, $\overline{X}_i$ is the mean of group $i$, $\overline{X}_i$ is the overall mean, and $N$ is the total number of observations [58]. A large F-ratio indicates that the between-group variance is significantly higher than the within-group variance. This suggests that the feature in question has a distinct distribution across different classes, which is essential for identifying informative genes in complex datasets like gene expression profiles.

**3.6.8. KL divergence.** KL Divergence It is also termed Kullback-Leibler Divergence. It measures how one probability distribution deviates from a reference distribution; hence, it is an essential measure for distinguishing features with distributional differences across classes in the context of feature selection. The usefulness of KL divergence in gene expression analysis: The features or genes whose expression distributions are disparate in the groups being examined, such as differences between cancer versus non-cancer samples, may reflect a difference in biological processes or specific disease expression patterns [59]. Mathematically, KL divergence from a distribution P (true distribution) to a distribution Q (reference distribution) is expressed as:

$$D_{KL}(P||Q) = \sum_i P(i) log \frac{P(i)}{Q(i)} \tag{11}$$

$P(i)$ and $Q(i)$ represent the probability of the outcome $i$ under distributions $P$ and $Q$, respectively. The sum extends over all possible outcomes, and KL divergence provides a non-symmetric measure of the information lost when $Q$ is used to approximate $P$ [60,61]. For gene expression data, features with high KL divergence values have distributions that diverge significantly between classes, making them informative candidates for classification tasks or biological interpretation.

**3.6.9. Variance threshold.** Variance Threshold is a straightforward feature selection technique that removes features with low variance, assuming that low-variance features contribute minimal information for distinguishing between classes. In gene expression data, for example, features (genes) that show slight variation across samples are likely to be uninformative, as they remain constant or nearly constant regardless of sample type [62]. Removing these features simplifies the dataset, reduces noise, and can improve model efficiency without sacrificing classification performance. Mathematically, for a feature $X_j$ with $n$ observations $X_{ji}$, $X_{j2}$, ….., $X_{jn}$ the variance $\sigma_j^2$ is calculated as

$$\sigma_j^2 = \frac{1}{n} \sum_{i=1}^{n} (X_{ji} - \overline{X}_j)^2 \tag{12}$$

$\overline{X}_j$ Is the mean of the feature $X_j$ across all samples. If $\sigma_j^2$ is below a predefined threshold, the feature is considered low variance and is removed from the dataset [63,64]. This technique effectively filters out features that do not contribute meaningful variance, retaining only those that show sufficient variability across samples for downstream analysis.

**3.6.10. Select from model.** Select From Model is a feature selection method that leverages feature importance scores from model-based estimators, such as Lasso or Random Forest, to retain only the most predictive features for a given dataset. This approach is highly adaptable, as it allows the selection of various estimators suited to different data types and classification tasks. For instance, Lasso regression uses L1 regularization, which assigns some coefficients a

zero value, effectively removing non-informative features. In contrast, ensemble methods like Random Forest evaluate importance of features by calculating the decrease in node impurity, enabling the identification of features that are strongly correlated with the target variable [65]. SelectFromModel reduces dimensionality, simplifies data representation, and enhances interpretability without sacrificing model accuracy by focusing on the most predictive features. Mathematically, SelectFromModel relies on feature importance scores. $Imp_j$ from a chosen estimator. For example, in Random Forest, the feature importance $Imp_j$ of a feature $X_j$ is often calculated based on the mean reduction in Gini impurity or entropy across all trees:

$$Imp_j = \frac{1}{T}\sum_{t=1}^{T}\Delta Imp_j^{(t)}$$

(13)

$T$ is the total number of trees and $\Delta Imp_j^{(t)}$ represents the reduction in impurity when $X_j$ is used for splitting in tree $t$. For feature selection, SelectFromModel ranks features by $Imp_j$ and discards those with importance scores below a specified threshold [66]. In applications like gene expression analysis, where many features may be irrelevant, SelectFromModel identifies a compact, informative subset of features, reducing computational burden and improving model performance.

**3.6.11. PCA and MI.** The PCA-MI hybrid approach is a powerful feature selection technique that combines PCA with MI to produce a feature set optimized for machine learning models, such as CNNs. This approach leverages PCA's ability to capture the main patterns of variance in the data and MI's focus on selecting features that are most relevant to the target. variable [42]. This results in a compact and informative feature set that retains essential global and class-specific information, especially for high-dimensional data like gene expression profiles. In the first step, PCA is applied to the dataset to reduce its dimensionality while retaining the directions of the largest variance. If X represents the original dataset with n samples and p features, PCA works by computing the covariance matrix C of X:

$$C = \frac{1}{n-1}X^TX$$

(14)

$X^T$ is the transpose of $X$. By calculating the eigenvalues and eigenvectors of $C$, PCA identifies the principal components, the eigenvectors associated with the largest eigenvalues. These principal components are ordered by the amount of variance they explain, with the first principal component capturing the most variance, followed by the second, and so on. If we denote the eigenvectors by $v_1, v_2, \ldots, v_k$ their corresponding eigenvalues by $\lambda_1, \lambda_2, \ldots, \lambda_k$ then, the principal components are chosen based on those eigenvectors with the largest eigenvalues, representing directions with maximum variance. For dimensionality reduction, we retain only the top $d$ principal components, reducing the dataset to a new matrix. $X_{PCA}$ of reduced dimensions:

$$X_{PCA} = X. \; [v_1, \; v_2, \; \ldots., v_d]$$

(15)

$d$ is selected to capture a high percentage (e.g., 95%) of the total variance, ensuring that the most significant patterns are retained. After PCA reduces the feature space, MI is applied to refine the feature set based on its relevance to the target variable. Mutual Information measures the dependency between each feature and the target, quantifying how much information about the target variable Y is gained by knowing the feature $X_i$ for each feature $X_i$ in $X_{PCA}$ and target $Y$, MI is calculated as

$$I(X; Y) = \sum_{x \in X}\sum_{y \varepsilon Y} p(x,y) log \frac{p(x,y)}{p(x)p(y)}$$

(16)

Features in $X_{PCA}$ with high MI scores are retained as they provide the most helpful information for predicting the target class, while those with low scores are discarded. This selection process results in a refined feature set PCA-MI that combines PCA's ability to capture overall data structure with MI's emphasis on class-specific relevance. This PCA-MI hybrid approach optimizes the feature space for the CNN model by retaining both global patterns and class-specific information, which can significantly improve classification performance. Reducing dimensionality with PCA and refining relevant features with MI enables the CNN to focus on informative features, resulting in more efficient training and improved predictive accuracy, particularly in complex, high-dimensional datasets like gene expression profiles for cancer classification. Each technique was applied independently to the dataset, and features extracted through each method were used to train a CNN classifier. Performance was measured using accuracy, precision, recall, and AUC scores, providing a comprehensive comparison [67]. The results demonstrated the PCA-MI hybrid method's superior performance, showcasing its ability to balance dimensionality reduction and feature relevance for effective lung cancer classification.

### 3.7. Protein-protein interaction analysis and Hub gene identification

To explore the biological significance of the key genes identified through our hybrid PCA-MI feature extraction framework, we conducted a detailed PPI analysis along with hub gene identification. These essential genes, derived from the hybrid framework's dimensionality reduction and relevance selection, were subjected to further analysis using the STRING database (version 12.0, available at https://string-db.org/). This database provides a platform for constructing high-confidence interaction networks by mapping genes to known and predicted protein-protein interactions. We focused on the human-specific protein interaction network, applying a stringent confidence score cut-off of ≥0.7. This threshold ensured the inclusion of reliable and biologically meaningful interactions, minimizing noise [68]. The resulting network comprised nodes (representing genes or proteins) and edges (denoting their interactions), effectively capturing the complex relationships among the significant genes. Once the network was constructed, it was exported from STRING and further analysed using Cytoscape (version 3.10.3), a widely used tool for visualising and analysing molecular interaction networks. Within Cytoscape, we employed the CytoHubba plugin to identify the central genes, or "hubs," in the PPI network. CytoHubba applies various centrality measures to rank nodes based on their importance or influence in the network. For our analysis, we used degree, betweenness, and closeness centrality measures to evaluate each gene's connectivity and regulatory potential [69]. The top 20 genes with the highest degree of connectivity were identified as hub genes, representing pivotal regulators within the network. These hub genes are propounded to play critical roles in lung cancer progression due to their extensive interactions and likely influence on key biological processes. Their identification provides valuable insights into the molecular mechanisms underlying lung cancer and highlights potential targets for therapeutic intervention or biomarker development. This integrative approach strengthens the biological relevance of the hybrid PCA-MI framework and underscores its utility in cancer research [70,71].

## 4. Results and discussion

### 4.1. Dataset

The dataset used for this experiment is curated by merging the gene expression data of the TCGA and ICGC repositories. The final benchmark dataset comprises samples from both sources, consisting of lung cancer subtypes and normal samples, which are diverse. TCGA contributed 1,153 samples comprising 541 adenocarcinoma samples, 502 squamous cell carcinoma samples and 110 normal samples, while ICGC contributed 543 samples comprising 488 adenocarcinoma samples and 55 normal samples. We used data from both prominent sources to achieve a broad dataset of high biological variability, which would be the basis of robust classification models [72].

**4.1.1. Data summary and statistics.** We have applied Z-score normalization on two datasets to ensure the compatibility of TCGA and ICGC samples. After normalization, we merged the two datasets based on common genes. We kept just those found in both datasets to form a coherent dataset that could be analysed reliably to pattern gene expression across the different types of cancer [73]. We, therefore, used mean imputation to smoothen out any missing

values so that our dataset would have no missing or gap spots. Once pre-processing was done, SMOTE came in handy in class imbalance. This includes using artificially generated synthetic samples from the minority class data. For this, normal samples were developed. This improves the classification model's credibility, especially for poorly represented classes [74].

| Source | Total Samples | Cancer Type | Sample Count |
|---|---|---|---|
| **TCGA** | 1,153 | Adenocarcinoma | 541 |
| | | Squamous Cell Carcinoma | 502 |
| | | Normal | 110 |
| **ICGC** | 543 | Adenocarcinoma | 488 |
| | | Normal | 55 |
| **Merged** | 1,696 | Adenocarcinoma | 1029 |
| | | Squamous Cell Carcinoma | 502 |
| | | Normal | 165 |

During the data preparation step, preprocessing and class balancing steps reduced potential biases and platform-dependent variations. Z-score normalization ensured a harmonization of data distributions for TCGA and ICGC samples, reducing differences from different sequencing protocols. Mean imputation further removed missing values without causing any form of interruption during analysis. Class balancing produced by SMOTE also eliminated an initially imbalanced nature of the dataset, offering a balance between normal and cancerous samples. The methodology proposed by increasing the number of underrepresented samples has prevented bias towards majority classes and supported the robustness of subsequent classification analysis [74].

## 4.2. Proposed hybrid model

The model proposes a novel hybrid approach for feature selection, combining PCA and Mutual Information to optimize the process, using a CNN model on lung cancer gene expression data for classification purposes. They both take advantage of the complements of PCA and MI: PCA reduces dimensions while MI evaluates feature relevance. Combined with both of these techniques, a lean, information-intensive feature set makes this CNN model better at properly classifying samples in lung cancer cases. The majority of the gene expression data have many high-dimensional features, including many redundant and irrelevant features not obviously relevant to classification. This can potentially degrade the model's performance with increased computational load and chances of overfitting, where more noise is imprinted on the model rather than meaningful patterns [42,75]. Traditional feature selection methods are effective as they either focus on reducing dimensionality, like PCA, or capturing the relevance of features, like MI, separately. The hybrid approach, combining PCA and MI, gives a comprehensive feature selection process that balances the dimensionality reduction with selecting features most relevant to distinguish cancer and non-cancer cases. The hybrid approach begins by using PCA to capture the principal components, explaining more than 95% of the variance in the data. According to the idea of variances, the feature is significantly reduced, yet the main underlying patterns in the data are kept. The application of MI, after PCA, ranks the features by mutual dependence concerning target labels such that those retained provide meaningful contributions toward the classification task. The two-step approach reduces dimensionality robustly without impairing classification potential. Significantly, classification accuracy and computational efficiency improved using the PCA-MI hybrid model. The CNN model trained on PCA-MI selected features achieved high predictive performance without increased computational cost by reducing feature sets to the most relevant ones. Hybrid outperforms individual feature selection methods: The results indicate that hybrid approaches have higher accuracy and, importantly, more excellent stability when evaluated across validation sets compared with particular approaches. This represents the hybrid model of PCA-MI as an efficient and effective feature selection approach in high-dimensional gene expression data. It involves PCA and MI to develop a feature set that can simultaneously reduce dimensionality and improve classification accuracy by retaining only the most

informative features [76,77]. The results of a CNN classifier trained on the optimized feature set provide excellent proof of the utility of this hybrid model in diagnosing lung cancer.

**4.3.Algorithm description for hybrid PCA-MI feature selection**
  **Step 1:** Data Collection & Merging
    *Download TCGA and ICGC datasets*
    *Merge datasets based on common genes*
  **Step 2:** Preprocessing
    *Impute missing values in merged dataset*
    *Apply SMOTE to balance class distribution in the target variable*
  **Step 3:** Feature Preparation
    *Separate features (X) and target variable (y)*
    *Encode target variable labels as integers*
    *Convert target labels to one-hot encoding for CNN compatibility*
  **Step 4:** Data Splitting & Scaling
    *Split data into training and testing sets*
    *Standardise features using a scaler*
  **Step 5:** Hybrid Feature Extraction
    *Apply PCA to reduce dimensionality, retaining the most informative components*
    *Use Mutual Information to select top features from the PCA output*
    *Save list of selected features for reference*
  **Step 6:** Data Reshaping
    *Reshape features to format compatible with CNN (samples, features, 1)*
  **Step 7:** Model Building
    *Define CNN architecture with appropriate convolutional, pooling, and dense layers*
    *Compile model for multi-class classification with loss and metrics*
  **Step 8:** Training
    *Train model on training set with early stopping to prevent overfitting*
  **Step 9:** Evaluation
    *Evaluate model performance on test set*
    *Calculate accuracy, precision, recall, and F1 score*
    *Save performance metrics to a file named **model_metrics.csv** in the output directory.*
  **Step 10:** Visualization
    *Generate and save a confusion matrix to **confusion_matrix.png** in the output directory*
    *Plot and save training and validation curves for accuracy and loss to **accuracy_loss_curves.png***
    *Plot and save the ROC curve to **roc_curve.png** for classification performance assessment.*

## 4.5. Evaluation metrics

We evaluated the performance of our hybrid PCA-MI model in a set of classification metrics: accuracy, recall, precision, F1-score, and the area under the Receiver Operating Characteristic curve-AUC. These metrics express various aspects of the model's ability to predict based on the given classification, with the need for both sensitivity and specificity. The metrics selected here are to provide an all-around view of model performance, balancing between the need for general accuracy and measures which account both for false positives and false negatives. Its importance is most dramatically seen in false negatives in cases related to cancer diagnosis, where a cancerous case is misclassified as being non-cancerous and severe consequences may follow [78]. This is why we used metrics that measure the model's capability to correctly classify positive cancer cases as such, correct the negative ones as non-cancer, and balance precision with recall.

  ***Accuracy***: Accuracy evaluates the percentage of correctly classified instances (both cancerous and non-cancerous) out of the total samples.

$$Accuracy = \frac{True\ Postives + True\ Negatives}{Total\ Number\ of\ Samples}$$

(17)

In binary classification, accuracy provides a quick overview of the model's correctness. However, it may not fully capture performance, especially in cases where the dataset classes are imbalanced (e.g., a more significant number of non-cancerous than cancerous samples).

**Precision:** Precision quantifies the number of true positive classifications among all samples predicted as positive by the model.

$$Precision = \frac{True\ Postive}{True\ Postives + False\ Positives} \tag{18}$$

High precision indicates fewer false positives, meaning the model is less likely to classify non-cancerous samples as cancerous incorrectly. This is particularly relevant to avoid overdiagnosis or unnecessary follow-up procedures.

**Recall (Sensitivity):** Recall (also known as sensitivity) is the proportion of actual positive cases that the model correctly identifies as positive.

$$Recall = \frac{True\ Positives}{True\ Positives + False\ Negatives} \tag{19}$$

Recall is crucial as it represents the model's ability to detect cancerous cases. High recall minimizes false negatives, reducing the chance of missing cancer diagnoses, which is vital in clinical applications.

**F1-Score:** The F1-score is the harmonic mean of precision and recall, providing a single metric that balances both. It's instrumental when there's a trade-off between precision and recall.

$$F1 - Score = 2 \times \frac{Precision \times Recall}{Precision + Recall} \tag{20}$$

The F1-score is beneficial when precision and recall are equally important, emphasising a balance between them. For example, in cases where it's crucial to detect as many cancer cases as possible without significantly increasing false positives, the F1-score becomes a meaningful metric.

**Area Under the ROC Curve (AUC):** AUC checks the area under the Receiver Operating Characteristic (ROC) curve, plots the true positive rate against the false positive rate at different threshold levels. The AUC is a reliable metric that assesses the model's ability to differentiate between classes at various thresholds. An AUC of 1 indicates perfect classification, while an AUC of 0.5 implies the model performs no better than random guessing. [78]. AUC is valuable in providing an aggregate measure of performance across all classification thresholds, making it particularly useful for assessing the reliability of models in medical diagnoses, where optimal decision thresholds are vital.

| Metric | Formula | Importance in Cancer Diagnosis |
|---|---|---|
| Accuracy | $\frac{TP+TN}{TP+TN+FP+FN}$ | General measure of correctness. |
| Precision | $\frac{TP}{TP+FP}$ | Reduces false positives, lowering the risk of overdiagnosis. |
| Recall | $\frac{TP}{TP+FN}$ | Minimizes false negatives, which is critical for identifying cancer cases. |
| F1-Score | $2 \times \frac{Precision \times Recall}{Precision+Recall}$ | Balances precision and recall, suited for balanced evaluation. |
| AUC | Varies with ROC plot | Assesses discriminatory power across thresholds. |

### 4.5. CNN model performance on PCA-MI selected features

Our hybrid PCA-MI method was applied for feature selection, followed by training and testing the outcome of this feature subset by utilizing a CNN classifier. This section provides a detailed performance analysis of the CNN on classification metrics, thus showing that PCA-MI is a good feature selection technique. Various classification metrics, such as accuracy,

precision, F1-score, recall, and area under the ROC curve (AUC), were used to validate the performance of the CNN model. These measures give an overall view of the CNN functionality in distinguishing between carcinogenic and noncarcinogenic samples: The CNN achieved PCA-MI selected features accuracy of 98%, which reflects the robustness. As can be seen, the model can identify many cancerous and non-cancerous samples [79]. Good precision suggests that PCA-MI has successfully retained the amount of information the CNN needs, enabling good generalization over the data [80]. Precision for the CNN model was 98%, which reflects the robustness as can be seen; the model can identify many cancerous and non-cancerous samples. Good precision suggests that PCA-MI has successfully retained the amount of information the CNN needs, enabling good generalization over the data. The precision for the CNN model was 98%. This high precision score means the model predicts cancerous samples correctly, which is important in clinical diagnostics, as false positives must be kept to an absolute minimum to avoid interference. The CNN had an F1-score of 97%, thus operating on a level playing field in calculating true positives without producing false positives and negatives. The result demonstrates the strength of PCA-MI boosting feature relevance and overall model performance. PCA-MI Feature Selection Impact on CNN Performance [81].

## 4.6. Impact of PCA-MI feature selection on CNN's performance

The PCA-MI hybrid feature selection technique significantly impacted CNN's classification performance, particularly its ability to accurately differentiate between cancer classes. It reduced the dimensions while preserving the significant features in prediction. This made CNN focus on quality information that would get class separation right. The high recall and AUC values indicate that the model can effectively distinguish the samples as cancerous, which is crucial for correct diagnosis and elimination of false positives. PCA-MI feature selection provided fewer features, thus reducing the overall complexity of the CNN model [82,83]. Reducing the volume of data made it possible to achieve faster training efficiency, wherein the convergence of the model was achieved with similar performance. The smaller feature set also makes the model architecture less complex, thus less prone to overfitting and improving generalizability. For clinical real-world applications, both precision and efficiency are desired in models. The hybrid approach with PCA and MI optimally balances the requirement of removing irrelevant features to focus on the most informative ones. In that way, the CNN model provides a practical solution by offering improved precision and recall and decreasing the chances of false positives and negatives while maintaining clinically relevant classifications with much reliability [84]. Confusion matrix **Fig 4A** demonstrates the performance of the CNN model in lung cancer subtype classification. All the classes are classified with high accuracy, demonstrating that the hybrid PCA-MI feature selection technique is robust for such complicated classification. For class Adenocarcinoma (A), the model accurately classified 301 out of 314 samples with an accuracy of approximately 96.9%. There were only minor misclassifications, with 4 samples incorrectly labelled as Normal (N) and 9 as Squamous (S). The Normal (N) class achieved a perfect classification rate, with all 306 samples correctly identified, showcasing the model's exceptional sensitivity and specificity for non-cancerous cases. For the Squamous (S) class, the model correctly classified 297 out of 307 samples. However, 9 samples were classified under A and 1 under N [85]. These results show the model to be good in distinguishing cancer samples from non-cancerous samples especially between similar categories of cancer **Fig 4B** is the ROC curve for the ability of the model to classify normal versus lung cancer subtypes. AUCs for all classes are very high at over 0.98 values, hence making High values for these AUCs demonstrate the clear-cut discrimination capability of the model

between true and false positives, further consolidating the power of hybrid PCA-MI feature extraction in optimizing the input features to feed into the CNN. Training Accuracy vs. Validation Accuracy, as shown in **Fig 4C**. Reflects the learning of the CNN model at every training step. During epochs, accuracy for both training and validation datasets was consistently increasing, reflecting a monotonic learning curve [86]. The slight difference between the training and validation accuracies ensures that overfitting has been well controlled. There was stability provided by appropriate dropout rates, such as 0.3

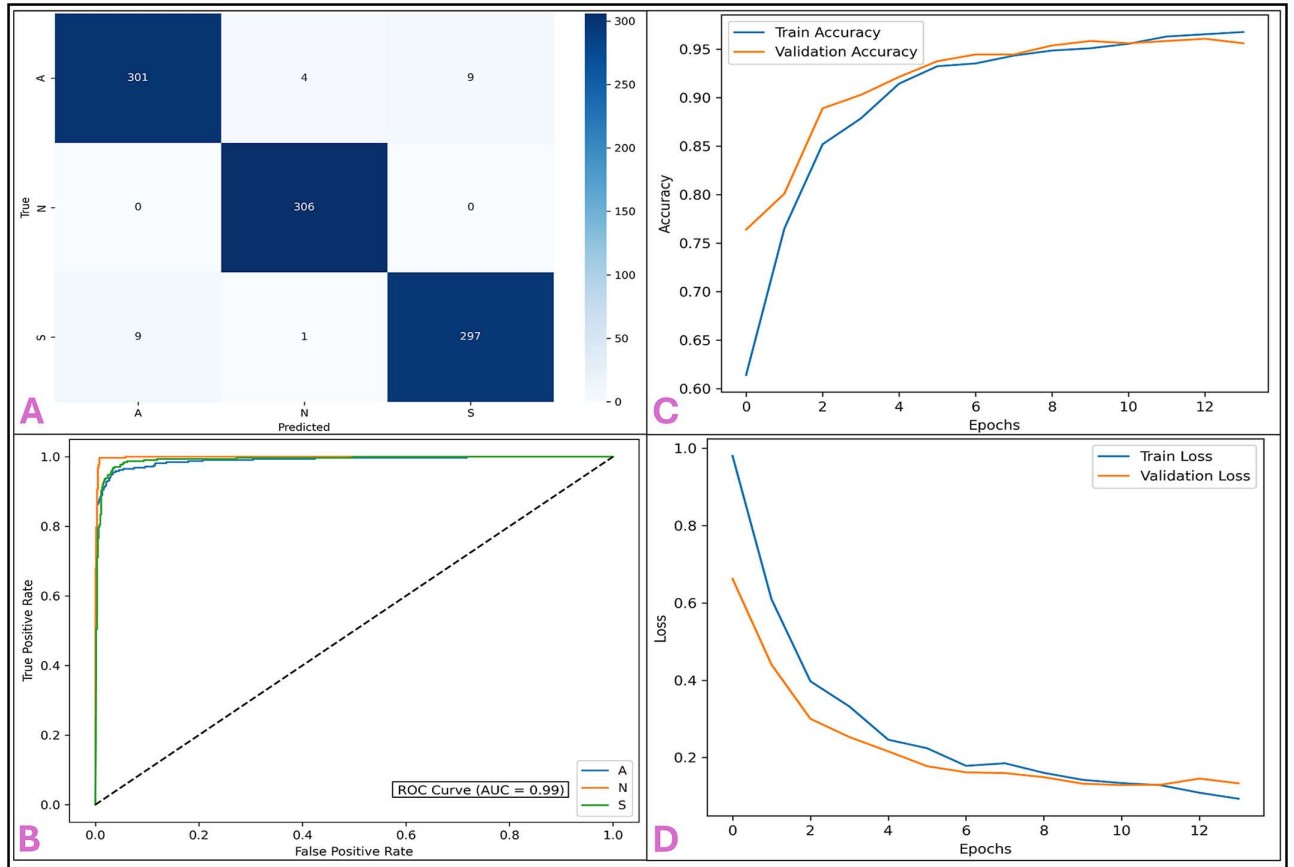

**Fig 4. Performance Evaluation of CNN Model, A) The confusion matrix highlights high classification accuracy, with minimal misclassifications among the Adenocarcinoma (A), Normal (N), and Squamous (S) classes. B**) The ROC curve shows an overall AUC of 0.99, indicating excellent discriminatory power across all classes. **C, D**) Training and validation accuracy and loss curves demonstrate consistent learning with minimal overfitting, achieving strong generalization and low error.

to 0.5, and the hybrid dependency on PCA-MI features that cut down on redundancy in input data, hence allowing better recognition patterns. The training and validation loss curves shown in **Fig 4D** are smooth and somewhat equitably decreasing for training and validation with successive epochs. The sharp drop in the training loss during the first epochs reflected strong learning of the features, and the validation loss curves track with a similar trend [87]. The balance between the shape of the two loss curves is far from overfitting, yet it still retains robust generalisation. These low final loss values prove the appropriateness of the chosen parameters of the training, especially the Adam optimiser with the ideal learning rate of 0.001, for obtaining optimal performance. The results prove the CNN model to be highly robust and reliable for the classification of lung cancer, along with high accuracy in the confusion matrix along with a high level ROC curve, signify that the model is highly efficient in differentiating between cancerous and non-cancerous cases as well as among different types of cancers. The stability of the training and validation metrics reflect a well-optimized and generalizable training process [88].

## 4.7. Comparative analysis with individual feature extraction techniques

The proposed PCA-MI hybrid feature selection approach is compared with several individual feature extraction techniques, and in detail, how each technique impacts the performance of the CNN classifier is discussed. This analysis is

very important to prove the effectiveness of the PCA-MI hybrid model compared to generally applied methods. To benchmark, we analyse the performance of the CNN model with respect to various known feature extraction techniques such as Autoencoder, PCA, Mutual Information, Autoencoder and Mutual Information, PCA-MI, Lasso, Random Forest, ANOVA, KL Divergence, Variance Threshold, Select from Model [79]. Each technique gives a different perspective on dimensionality reduction and feature selection in a specific aspect of the features' relevance, redundancy, and predictiveness.

**4.7.1**. **CNN performance metrics using each feature extraction technique.** We trained the CNN model for each extraction technique and reported results using all metrics above, including accuracy, precision, recall, F1-score, the area under the ROC curve, or simply AUC. These metrics provide a holistic view of the model's classification performance, reflecting its ability to identify cancerous samples and minimize misclassifications accurately. Summary of results Below is a summary of the results, which reveals that the PCA-MI hybrid model consistently outperformed individual feature extraction methods across most evaluation metrics, demonstrating its advantage in effectively balancing dimensionality reduction with feature relevance [79].The **Table 1**. shows that the hybrid model Combining PCA-MI surpassed individual feature extraction techniques on all the metrics evaluated. The superiority might be because PCA is a feature reduction technique that retains significant components, while MI focuses on retaining the most relevant features for prediction. Thus, PCA-MI reduces the heavy computational load, improving classification accuracy and good reliability. [79].It enhances both the recall and the F1-score, which indicates the detection of precise cancer cases without losing any specificity, which is highly needed in such medical applications. In this case, the individual techniques, effective in various ways or others, either fought with reducing the dimensionality, such as SelectFromModel and Laso failed to capture relevant features due to redundancy, as in Autoencoder and Random Forest.

The proposed hybrid method achieves the highest accuracy [79]. **Fig 5A**. Depicts the accuracy obtained by each feature extraction method in the form of a bar chart. The proposed hybrid method's accuracy is much higher than that of the other techniques: MI (0.93) and Autoencoder (0.92). This shows the necessity of developing a fusion between dimensionality reduction methods and feature selection to improve the classification model. **Fig 5B** shows the metrics of the proposed model [83]. The hybrid PCA-MI also further facilitated the retrieval of specific significant genes with respect to high feature importance scores. The key genes found include CYP51A1 (score: 0.459), TNMD (score: 0.776), and NFYA (score: 0.306), among others. These genes are critical in the classification process and relate more to the biological aspects of lung cancer diagnosis. The hybrid approach demonstrates a unique advantage in prioritizing genes that contribute most significantly to predictive accuracy. The method minimises redundancy and noise while preserving key biomarkers by leveraging PCA to reduce dimensionality and MI to retain relevant features [89,90]. This is especially vital

**Table 1. The Accuracy, Precision, Recall, and F1-Score for various feature extraction methods. The proposed method achieves the highest accuracy across metrics.**

| Feature Extraction Method | Accuracy | Precision | Recall | F1-Score |
|---|---|---|---|---|
| Preposed | 0.98 | 0.97 | 0.98 | 0.97 |
| MI | 0.93 | 0.94 | 0.94 | 0.93 |
| Autoencoder+MI | 0.94 | 0.95 | 0.94 | 0.94 |
| Autoencoder | 0.92 | 0.93 | 0.92 | 0.91 |
| Lasso | 0.95 | 0.96 | 0.94 | 0.95 |
| Random Forest | 0.96 | 0.95 | 0.96 | 0.95 |
| ANNOVA | 0.95 | 0.96 | 0.96 | 0.95 |
| SelectFromModel | 0.96 | 0.95 | 0.96 | 0.95 |
| KL Divergence | 0.94 | 0.93 | 0.94 | 0.94 |
| PCA | 0.93 | 0.94 | 0.93 | 0.93 |

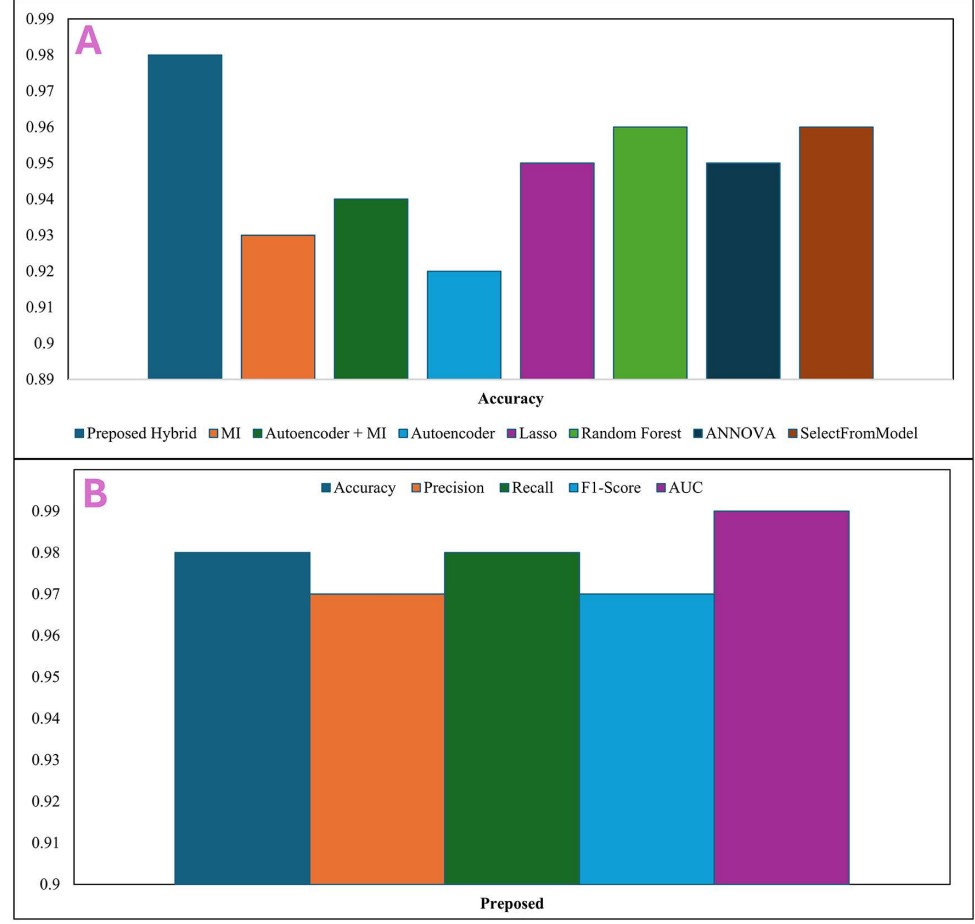

**Fig 5. This chart shows the accuracy of various feature extraction methods.**

in gene expression datasets, where understanding the biological relevance of features is critical for translational research. **Fig 6** presents gene scores in more detail, showing that PCA-MI could differentiate the most predictive genes from less relevant ones. Such insights can then be helpful in further biological validation and exploration of identified genes as potential biomarkers for lung cancer [91].Following identifying significant genes with the PCA-MI hybrid feature extraction framework, an in-depth Protein-Protein Interaction (PPI) analysis and hub gene identification were performed to provide biological significance of these key genes associated with lung cancer. PPI network analysis offers insights into complex molecular interactions of proteins encoded by the most significant genes selected with the feature extraction process. These genes were mapped into known and predicted protein-protein interactions within a human-specific network using the STRING database (version 12.0), applying a stringent confidence score cutoff of ≥0.7 [68,92]. This ensured we captured high confidence interactions but allowed sufficient entries for biological meaningfulness. The network was exported into Cytoscape (version 3.10.3) for further visualization and analysis. **Fig 7** shows the PPI network with proteins as nodes and edges for interaction. The complex interrelation seemed to indicate the central involvement of specific genes in the pathophysiology of lung cancer. To identify the most central or influential genes, we used the CytoHubba plugin of Cytoscape. Using centrality measures like degree, betweenness, and closeness, we ranked the genes with the highest connectivity and, thus, the most important for regulatory purposes [69].

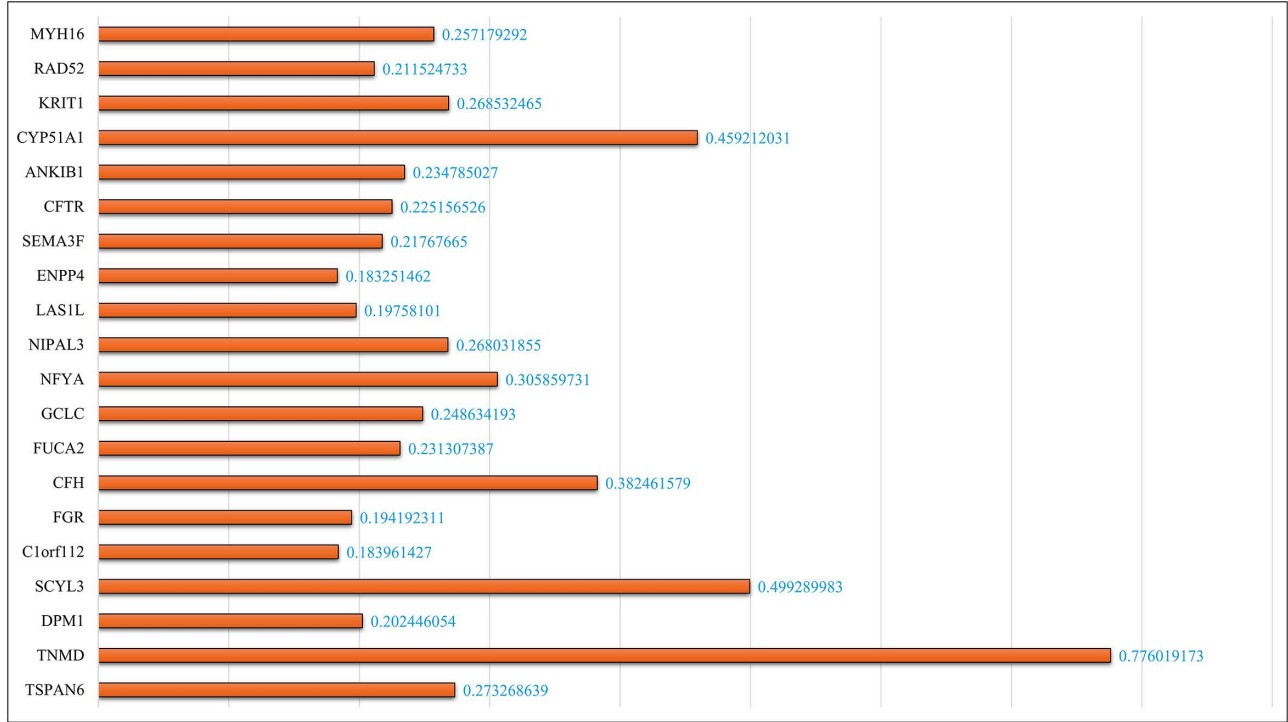

**Fig 6. Showcases the significance of the top genes and the importance score identified by the hybrid approach, validating its effectiveness for biomarker discovery.**

Degree centrality measures how many direct connections a node owns; betweenness centrality identifies nodes that can act as bridges linking clusters, and closeness centrality is an assessment of how near a gene or a node is to all other elements in the network. The top 20 hub genes found with this analysis were considered critical regulators within the network, and hence, they were of potential importance in lung cancer progression. Such genes include the critical regulators BRCA1, CD4, RAD51, and CFTR, significantly affecting cellular signalling, extracellular matrix remodelling, and tumour progression. **Fig 8** Results from hub gene analysis [93]. Part **A** of the Fig shows the top 20 hub genes selected in the CytoHubba plugin, focusing on the fact that these are central nodes in the PPI network. Gene nodes are colour-coded in a gradient from red to yellow; the darker the red, the more significant the connectivity. Part **B** of the Fig indicates the ranking of the hub genes regarding the degree of centrality: a quantitative way of representing their strength in the network. Having this dual representation of the importance of these genes helps us understand their roles in the molecular landscape of lung cancer. The use of PPI network analysis and hub gene identification points to the biological significance of the PCA-MI framework [94].

The identified hub genes could become novel biomarkers for diagnosing lung cancer or its progression and will be considered critical therapeutic targets. This approach validates the significance of the hybrid feature extraction method and bridges computational and biological insights, allowing a pathway for findings in translation to clinical applications. The whole analysis enhances the entire framework so much that it can be used in other cancers or high-dimensional datasets.

## 5. Limitation and future work

This hybrid feature extraction strategy shows significant promise, especially in the context of gene expression data, where high dimensionality poses a considerable challenge. However, there are limitations to this work. While the feature extraction techniques we employed enhance accuracy, they also add complexity, which may limit the generalisability of the final model. Additionally, this

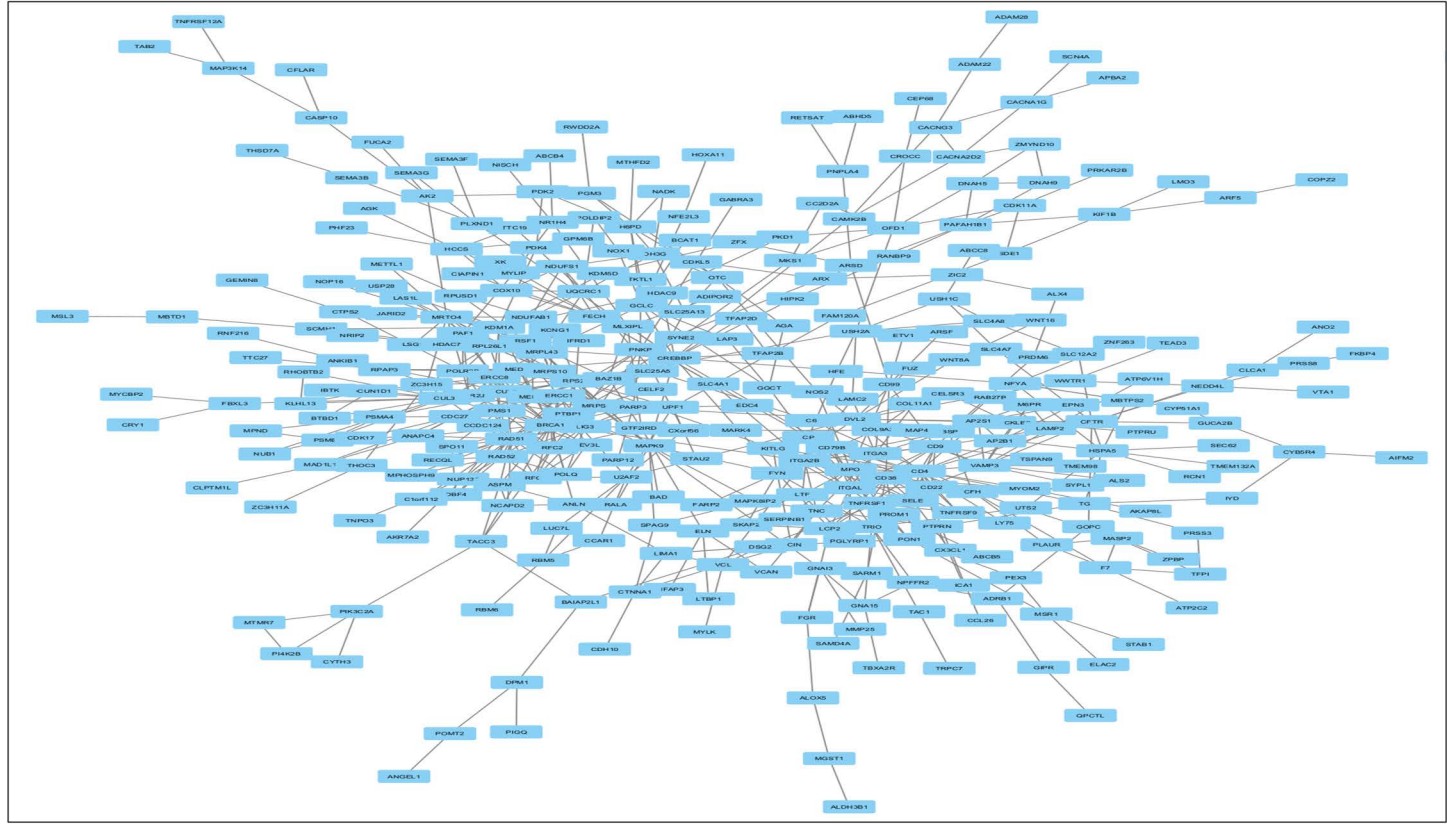

**Fig 7. The PPI network visualises interactions among significant genes identified through the PCA-MI framework, with nodes representing proteins and edges denoting interactions, constructed using STRING with a confidence score ≥0.7.**

study specifically focuses on lung cancer classification using gene expression profiles that may vary across populations and environments, potentially impacting the model's adaptability. We demonstrated the effectiveness of our approach; real-world validation in clinical settings would be necessary to confirm its practical applicability, a step that remains beyond the current scope of this study.

### 5.1. Future prospects

One potential avenue is integrating multi-omics data such as proteomics, metabolomics, and clinical data alongside gene expression data. This could provide a more holistic view of cancer progression and help identify specific and reliable biomarkers across different data types. Combining various data types may also allow us to capture biological complexity better and lead to more personalized diagnostic and treatment options. Another important future direction is validating this approach with larger, more diverse datasets that include a more comprehensive range of patient demographics and environmental factors. This would help assess the model's robustness and adaptability to real-world clinical scenarios. The hope is that such models will help support personalized medicine, allowing healthcare providers to tailor treatments and improve patient outcomes based on individual genetic profiles.

### 6. Conclusion

This study addresses a critical challenge in lung cancer diagnosis with a proposed hybrid feature extraction framework that effectively manages the dimensionality of gene expression data—a significant obstacle to accurate

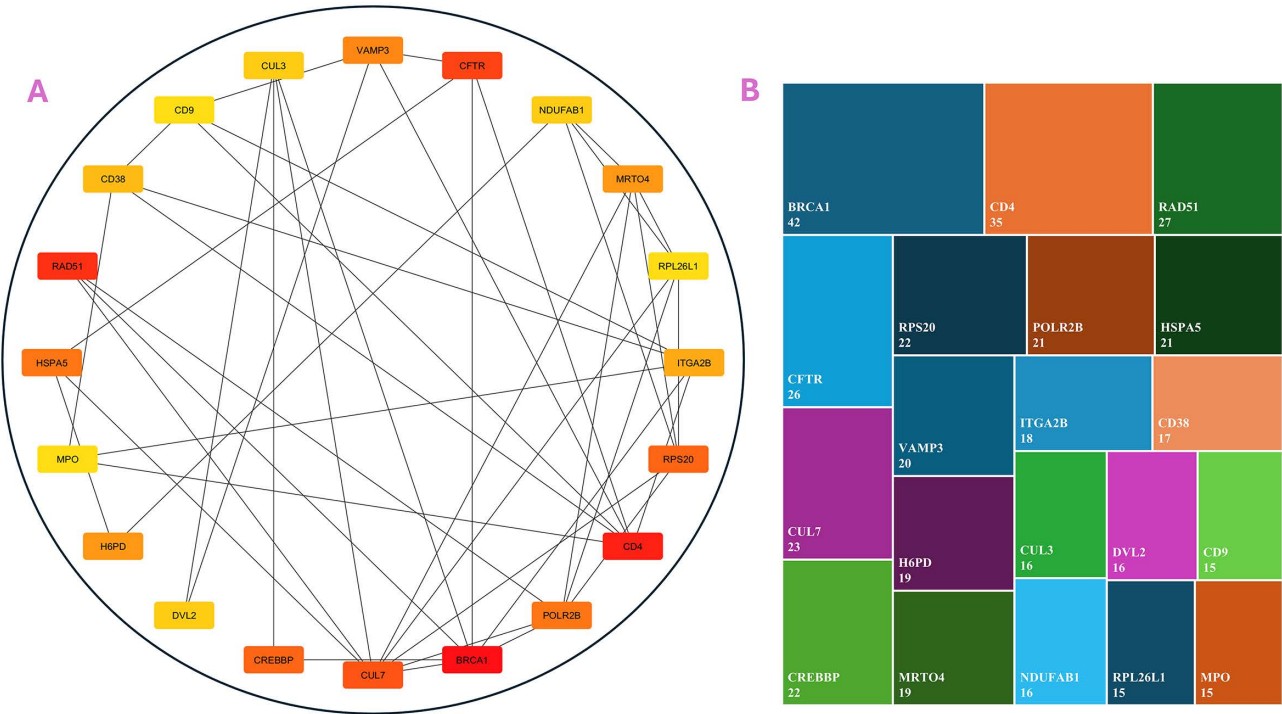

**Fig 8. Hub Gene Analysis: A) Top 20 hub genes ranked by degree centrality are highlighted, indicating their pivotal roles in the network. B)** chart showing degree centrality scores of the hub genes, reflecting their connectivity and influence.

classification. Lung cancer is one of the deadliest cancers globally, and early, accurate detection is crucial for improving survival rates. While gene expression data capture the biological mechanisms underlying lung cancer, the high dimensionality of these datasets often increases computational demands and risks of overfitting. To overcome these challenges, we developed a hybrid approach combining PCA and MI for feature extraction. PCA reduces dimensionality by retaining components that explain over 95% of the variance, focusing on critical patterns while filtering out noise. Concurrently, MI identifies features highly relevant to the target class, ensuring the feature set is concise and biologically informative. Using this approach, we created a benchmark dataset by merging gene expression data from TCGA and ICGC, based on shared genes, to provide a robust basis for our classification model. A CNN trained on the PCA-MI reduced dataset demonstrated high classification performance, achieving 98% accuracy and precision, underscoring its effectiveness in distinguishing lung cancer samples. Comparative analysis with ten other feature extraction methods, including Lasso, Random Forest, and Others, confirmed the superiority of the PCA-MI hybrid approach. Training and validation curves highlighted stable learning behaviour, and confusion matrix analysis validated the model's predictive accuracy. Additionally, genes ranked by the PCA-MI framework were analysed using PPI networks, identifying 20 hub genes like BRCA1, CD4, RAD51, and CFTR proposed to play pivotal roles in lung cancer biology. These findings reinforce the biological relevance of the selected features, bridging computational analysis with biological insights. This hybrid framework demonstrates the potential to form the foundation for advanced cancer diagnostic tools, particularly in multi-omics data integration, where managing large, complex datasets is critical. Future research could explore its application to other cancer types and the integration of additional data sources, such as proteomics and metabolomics, to further improve diagnostic accuracy and provide deeper biological insights.

## Author contributions

**Conceptualization:** Syed Naseer Ahmad Shah, Rafat Parveen.

**Data curation:** Syed Naseer Ahmad Shah, Rafat Parveen.

**Formal analysis:** Syed Naseer Ahmad Shah, Kaartik Issar.

**Funding acquisition:** Kaartik Issar.

**Investigation:** Syed Naseer Ahmad Shah.

**Methodology:** Syed Naseer Ahmad Shah, Kaartik Issar.

**Project administration:** Syed Naseer Ahmad Shah, Kaartik Issar.

**Resources:** Syed Naseer Ahmad Shah, Rafat Parveen.

**Software:** Rafat Parveen.

**Supervision:** Rafat Parveen.

**Validation:** Syed Naseer Ahmad Shah, Kaartik Issar.

**Visualization:** Syed Naseer Ahmad Shah, Kaartik Issar.

**Writing – original draft:** Syed Naseer Ahmad Shah, Kaartik Issar, Rafat Parveen.

**Writing – review & editing:** Syed Naseer Ahmad Shah, Kaartik Issar, Rafat Parveen.

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
