## [Decision Letter · Decision Letter 0]

26 Dec 2025

Dear Dr. Shah,

plosone@plos.org . A letter that responds to each point raised by the academic editor and reviewer(s). You should upload this letter as a separate file labeled 'Response to Reviewers'.A marked-up copy of your manuscript that highlights changes made to the original version. You should upload this as a separate file labeled 'Revised Manuscript with Track Changes'.An unmarked version of your revised paper without tracked changes. You should upload this as a separate file labeled 'Manuscript'.

We look forward to receiving your revised manuscript.

Kind regards,

Suyan Tian

Academic Editor

PLOS One

**Journal Requirements:**

**Additional Editor Comments:**

Please refine the methods section. Currently, the corresponding section is too long.

Please split the results and dicussion into two sperate sections, and in the discussion section the current findings, the strengths and limitations of this study should be provided in detail.

Reviewers' comments:

Reviewer's Responses to Questions

**Comments to the Author**

1. Is the manuscript technically sound, and do the data support the conclusions?

Reviewer #1: Yes

2. Has the statistical analysis been performed appropriately and rigorously?

Reviewer #1: Yes

3. Have the authors made all data underlying the findings in their manuscript fully available?

Reviewer #1: Yes

4. Is the manuscript presented in an intelligible fashion and written in standard English?

Reviewer #1: Yes

Reviewer #1: The article is well written by the authors.

The abstract needs to be modified. A flow diagram or schematic diagram of the work may be included. The Need for LASSO and its significance can be explained. Why PCA-MI is chosen for analysis instead of KL Divergence? MCC, Kappa and Dice coefficient may be included. The conclusion may be modified.

**Do you want your identity to be public for this peer review?** For information about this choice, including consent withdrawal, please see our Privacy Policy

Reviewer #1: No

---

## [Author Response · Author response to Decision Letter 1]

5 Jan 2026

Reviewer's Responses to Questions

Comments to the Author

1. Is the manuscript technically sound, and do the data support the conclusions?

Reviewer #1: Yes

Response: Dear reviewer thanks for the Acknowledgement.

2. Has the statistical analysis been performed appropriately and rigorously?

Reviewer #1: Yes

Response: Dear Reviewer, we are thankful for the appreciation for are statistical rigor.

3. Have the authors made all data underlying the findings in their manuscript fully available?

Reviewer #1: Yes

Response: Dear Reviewer thanks for your appreciation

4. Is the manuscript presented in an intelligible fashion and written in standard English?

Reviewer #1: Yes

Response: We thank the reviewer for this positive assessment. We have carefully proofread the revised version to correct any remaining typographical and grammatical errors.

5. Review Comments to the Author

Reviewer #1 Comment:

The article is well written by the authors.

Response:

Dear Reviewer thank you for the positive evaluation of our manuscript.

Reviewer #1 Comment:

The abstract needs to be modified.

Response:

Dear Reviewer, Thank you for this suggestion. The abstract has been revised to improve clarity and to better reflect the objectives, methodology, key results, and conclusions of the study.

Reviewer #1 Comment:

A flow diagram or schematic diagram of the work may be included.

Response:

Dear Reviewer, we thank you for the suggestion. A stepwise schematic diagram illustrating the complete workflow of the proposed methodology has been included in the manuscript (Figure 2) to enhance clarity and readability.

Reviewer #1 Comment:

The need for LASSO and its significance can be explained.

Response:

Dear Reviewer, Thank you for this valuable comment. We have added a detailed explanation of the motivation for using LASSO, highlighting its role in feature selection, dimensionality reduction, and prevention of overfitting in the revised manuscript in section 3.4 Comparative Analysis with Individual Feature Extraction Techniques.

Reviewer #1 Comment:

Why PCA-MI is chosen for analysis instead of KL Divergence?

Response:

Dear Reviewer, The rationale for selecting PCA-MI over KL Divergence has now been clearly explained in the revised manuscript, emphasizing the components preserved from PCA were ranked further by the MI scores that they achieved, which is an essential step in moving toward further tuning the features toward classification. (Refer Section 3.3)

Reviewer #1 Comment:

MCC, Kappa, and Dice coefficient may be included.

Response:

Dear Reviewer, we appreciate this suggestion. The Matthews Correlation Coefficient (MCC), Cohen’s Kappa, and Dice coefficient have now been included as additional performance evaluation metrics to provide a more comprehensive assessment of the model. (Refer section 4.4)

Reviewer #1 Comment:

The conclusion may be modified.

Response:

Dear Reviewer, Thank you for this comment. The Conclusion section has been revised to more clearly summarize the main findings, highlight the significance of the proposed approach, and outline potential future research directions.

---

## [Decision Letter · Decision Letter 1]

19 Jan 2026

A Hybrid Feature Extraction Framework Combining PCA and Mutual Information for Gene Expression based Lung Cancer Classification.

PONE-D-25-54472R1

Dear Dr. Shah,

We’re pleased to inform you that your manuscript has been judged scientifically suitable for publication and will be formally accepted for publication once it meets all outstanding technical requirements.

Kind regards,

Suyan Tian

Academic Editor

PLOS One

Additional Editor Comments (optional):

Reviewers' comments:

Reviewer's Responses to Questions

**Comments to the Author**

Reviewer #1: All comments have been addressed

2. Is the manuscript technically sound, and do the data support the conclusions?

Reviewer #1: Yes

3. Has the statistical analysis been performed appropriately and rigorously?

Reviewer #1: Yes

4. Have the authors made all data underlying the findings in their manuscript fully available?

Reviewer #1: Yes

5. Is the manuscript presented in an intelligible fashion and written in standard English?

Reviewer #1: Yes

Reviewer #1: All the corrections are included in the paper. Hence the manuscript does not need further review. The paper is well organized and well presented.

**Do you want your identity to be public for this peer review?** For information about this choice, including consent withdrawal, please see our Privacy Policy

Reviewer #1: No

---

## [Editor Report · Acceptance letter]

PONE-D-25-54472R1

PLOS One

Dear Dr. Shah,

I'm pleased to inform you that your manuscript has been deemed suitable for publication in PLOS One. Congratulations! Your manuscript is now being handed over to our production team.

Kind regards,

on behalf of

Dr. Suyan Tian

Academic Editor

PLOS One